# Monitoring Welfare of Individual Broiler Chickens Using Ultra-Wideband and Inertial Measurement Unit Wearables

**DOI:** 10.3390/s25030811

**Published:** 2025-01-29

**Authors:** Imad Khan, Daniel Peralta, Jaron Fontaine, Patricia Soster de Carvalho, Ana Martos Martinez-Caja, Gunther Antonissen, Frank Tuyttens, Eli De Poorter

**Affiliations:** 1Department of Pathobiology, Pharmacology and Zoological Medicine, Faculty of Veterinary Medicine, Ghent University, 9820 Merelbeke, Belgium; imad.khan@ugent.be (I.K.); patricia.sosterdecarvalho@ugent.be (P.S.d.C.); anamartosmc@gmail.com (A.M.M.-C.); gunther.antonissen@ugent.be (G.A.); 2Department of Computer Science and Artificial Intelligence, University of Granada, 18071 Granada, Spain; 3DaSCI Andalusian Institute in Data Science and Computational Intelligence,18071 Granada, Spain; 4IDLab, Department of Information Technology, Ghent University—imec, 9052 Ghent, Belgium; jaron.fontaine@ugent.be (J.F.); eli.depoorter@ugent.be (E.D.P.); 5Poulpharm, 8870 Izegem, Belgium; 6Department of Veterinary and Biosciences, Faculty of Veterinary Medicine, Ghent University, 9820 Merelbeke, Belgium; frank.tuyttens@ugent.be; 7Flanders Research Institute for Agriculture, Fisheries, and Food (ILVO), 9090 Melle, Belgium

**Keywords:** wearable, animal welfare, broilers, chickens, UWB, IMU, accelerometer

## Abstract

Monitoring animal welfare on farms and in research settings is attracting increasing interest, both for ethical reasons and for improving productivity through the early detection of stress or diseases. In contrast to video-based monitoring, which requires good light conditions and has difficulty tracking specific animals, recent advances in the miniaturization of wearable devices allow for the collection of acceleration and location data to track individual animal behavior. However, for broilers, there are several challenges to address when using wearables, such as coping with (i) the large numbers of chickens in commercial farms,(ii)the impact of their rapid growth, and (iii) the small weights that the devices must have to be carried by the chickens without any impact on their health or behavior. To this end, this paper describes a pilot study in which chickens were fitted with devices containing an Inertial Measurement Unit (IMU) and an Ultra-Wideband (UWB) sensor. To establish guidelines for practitioners who want to monitor broiler welfare and activity at different scales, we first compare the attachment methods of the wearables to the broiler chickens, taking into account their effectiveness (in terms of retention time) and their impact on the broiler’s welfare. Then, we establish the technical requirements to carry out such a study, and the challenges that may arise. This analysis involves aspects such as noise estimation, synergy between UWB and IMU, and the measurement of activity levels based on the monitoring of chicken activity. We show that IMU data can be used for detecting activity level differences between individual animals and environmental conditions. UWB data can be used to monitor the positions and movement patterns of up to 200 animals simultaneously with an accuracy of less than 20 cm. We also show that the accuracy depends on installation aspects and that errors are larger at the borders of the monitored area. Attachment with sutures had the longest mean retention of 19.5 days, whereas eyelash glue had the shortest mean retention of 3 days. To conclude the paper, we identify current challenges and future research lines in the field.

## 1. Introduction

Monitoring the behavior of broiler chickens is essential for ensuring their welfare, detecting potential risks and taking timely remedial measures. Traditional manual observation methods are generally labor-intensive, do not provide continuous measurements over extended periods, and are prone to observer bias [1]. To address these limitations, sensor technologies are increasingly being employed to measure behaviors and welfare indicators of individual animals. Recent research has demonstrated the efficacy of various individual animal-based sensors in monitoring chicken behavior and activity levels.

IMU wearables have been used on broilers for the continuous monitoring of activity levels [2,3,4,5], sickness detection [6], poison detection [7], and activity classification [8,9,10]. Other studies have used this type of sensor on laying hens [11,12], but not yet on broilers.

Similarly, wearable Ultra-Wideband (UWB) systems have proven effective in tracking the position, movements, and behaviors of poultry [13], including fast-growing broilers in commercial conditions [14]. This type of tracking system has also been used to estimate the distance walked by chickens, although it was also shown that accumulated errors cause an overestimation [15]. This research line shows a lot of promise, but the amount of data gathered has usually been restricted to a few minutes or hours per chicken; we still lack studies that span several weeks. Furthermore, there now exist devices that integrate IMUs and UWBs into the same tag with a shared battery, minimizing the weight and impact in comparison to attaching both sensors separately and opening new possibilities for research and application in commercial farms.

Various attachment methods are currently employed for attaching sensors to poultry. Backpack harnesses, primarily used around the wings, allow sensors to be fixed on the birds’ backs [13,14]. Smaller sensors like Radio Frequency Identification (RFID) tags are often attached to legs using elastic bands [16]. Non-stretching sport tape has also been used to affix sensor nodes to the abdominal skin of chickens [17]. However, attaching on-body sensors to broilers has its challenges. The rapid growth of broilers requires frequent readjustment of backpack harnesses, leading to increased handling and potential stress. Moreover, the combined weight of the sensor and attachment method should be minimized, with a recommended maximum of 3–5% of the bird’s body weight [18,19]. Additional concerns include potential discomfort, wing damage, and premature detachment of sensors. Despite the growing use of sensor technologies in poultry research, there is a lack of comprehensive studies comparing different attachment methods specifically for broilers. The body structure and rapid growth of broilers warrant a targeted approach to sensor attachment that balances data collection needs with animal welfare concerns.

It is clear that long-term studies on optimal attachment methods and long-term IMU and UWB sensor findings, especially for broilers, are still lacking in the scientific literature. This pilot study aimed to address this knowledge gap by testing various methods of sensor attachment to broilers, the optimal settings of the IMU and UWB sensors, the potential lifetime of a monitoring system, and trade-offs in accuracy and weight. Our primary objective was to identify methods that provide durable attachment from as young an age as possible, minimize stress to the birds, and do not alter their natural behaviors. By exploring alternative attachment methods, this study aimed to improve the reliability and reduce the welfare implications of sensor-based monitoring in broiler research, potentially leading to more effective and ethical data collection practices in poultry science. Furthermore, the UWB and IMU sensors attached to the broilers were tested to measure noise and uncertainty, as well as the synergy between the two sensors, to establish technical guidelines that can be followed by future practitioners in industry or research. To the best of our knowledge, this is the first study to monitor broilers using wearable devices for a large part of their lifetime (starting from the age of 3 weeks, when they can carry the weight of the sensor), and also the first to incorporate both types of sensors simultaneously on broilers.

The remainder of this paper is structured as follows. First, Section 2 describes common technologies for tracking individual animals and attachment methods used for attaching wearables to chickens. Section 3 describes the settings of the experiments carried out. Section 4 describes the attachment methods tested and the results obtained in the experiments. Section 5 and Section 6 describe the experiments carried out to evaluate the UWB and IMU, respectively, and their results. Next, Section 7 provides practical guidelines on how to combine both sensors in a single wearable device. Section 8 provides a discussion of the findings of the previous sections. Finally, Section 9 concludes this paper by summarizing the main findings of our experimental study, establishing guidelines for practitioners, and identifying future lines of research in the field.

## 2. Related Work

### 2.1. Wireless Sensors: UWB and IMU

Since satellite communication typically does not penetrate buildings, Ultra-Wideband (UWB) is a technology that can be used as an indoor replacement for GPS localization. Instead of using satellites, localization with UWB requires the installation of a fixed number of UWB anchors with known locations in the space where the localization is carried out, such as a building or an outdoor enclosure. These act as virtual “satellites”, allowing us to localize any wearable that also has a UWB receiver or tag. Compared to GPS positioning, UWB can achieve much higher accuracy, with errors <20 cm, due to its very high temporal resolution, combined with algorithmic improvements (i.e., machine learning error correction, anchor selection, particle and Kalman filters, etc.) [20].

UWB localization implies deploying one or more anchors along with the wearables that are localized. This localization can be realized in two main ways. Two-way ranging (TWR) estimates the location by measuring the time taken by the signal to travel between the tag and each anchor. The distance between the tag and each anchor node is calculated sequentially. This approach is typically very accurate, as it eliminates the need for synchronization between multiple anchors and enables two-way communication between the tag and anchor. However, it requires the transmission of more packets (one packet to each anchor node), making it less suitable for high-density, low-energy use cases. In contrast, time difference of arrival (TDOA) involves several anchors simultaneously, requiring precise sub-nanosecond synchronization between them, but providing accurate positioning after sending only a single UWB packet from the tag. This reduces the number of required packets and offers greater scalability with a lower energy footprint [21], making TDOA the most suitable positioning protocol for applications like broiler monitoring. To this end, we chose UWB TDOA for the positioning experiments that were analyzed in this study.

Radio Frequency Identification (RFID) technology also enables localization with low battery consumption, although with a lower precision than UWB, as it is restricted to detecting which antenna is the closest to the wearable tag. Therefore, RFID deployment in a broiler pen with reasonable precision would require multiple antennae, which is why a single study has used this technology on broilers so far [22].

A second common technology used in wearables to analyze animal behavior is Inertial Measurement Unit (IMU) sensors. These sensors usually incorporate accelerometers, which capture changes in movement, and gyroscopes, which measure orientation changes. Similarly to UWB, IMU sensors have been used to perform activity recognition in sports [23], pedestrian dead reckoning [24], gait analysis [25], etc. These applications benefit from high sampling rates (typically > 20 Hz) and high detection accuracy by facilitating intelligent algorithms and machine learning, even on embedded hardware [26].

### 2.2. Broiler Monitoring

Building on the success of both UWB and IMU in other research domains, sensors have similarly shown promise for effective broiler monitoring. In this context, the first question that should be answered is whether the presence of these devices might alter the chicken’s behavior or welfare. Several studies found that the presence of wearable devices with a weight under 5% of the chicken’s body weight do not affect the chicken’s behavior, leg health, cleanliness, and production in the long term, although there is usually an adaptation period of 1–2 days after the device is fitted [13,27]. In [27], a cohort of broilers with and without harnesses were put together in the same pen to evaluate their interactions. The researchers observed that, in the first two days after attaching the harness, harnessed birds performed less exploration and locomotory behavior, and more aggression occurred. However, after this period, no behavioral difference was observed between harnessed and non-harnessed birds.

Studies using wearables can be categorized according to the technology used: RFID, UWB, and IMU [28]. Table 1 presents an overview of the studies that used wearable devices on poultry.

#### 2.2.1. IMU

Most of the previous applications of wearable devices on poultry in general, and broilers in particular, were carried out using IMUs, usually reading the accelerometer information.

The first applications of wearables on poultry were carried out on laying hens. Banerjee et al. [29] classified several activities using 2-D accelerometer data. Later, the same authors used accelerometers to detect jumps, the time of jump initiation and landing, the height of the jump, and the force of the landing [11]. Fujinami et al. [12] also used the information coming from a gyroscope to classify 12 different behaviors. The authors tested an LSTM model, but its results were worse than those from manually engineered features combined with an LGBM classifier.

The study with the largest number of broilers conducted so far (280) was [4], which presented a statistical analysis of bird behavior. Accelerometers were used to measure the inactivity of the birds. The study characterized the typical inactive time as 78–80% of the day.

In [8], an accelerometer was used to predict the presence of three behaviors (feeding/pecking, preening, and dust bathing) in a time window. Then, each of these classes was predicted as a binary classification problem using a sliding window that finds frequent motifs for said class. The activity frequency calculated for this dataset was further used to train a Conditional Tabular Generative Adversarial Network (CTGAN) [30] that generated more data to balance three classes (active, inactive, sick) and then identify sick chickens based on activity frequency but without using the accelerometer data [6].

Yang et al. [10] predicted four activities (walking, resting, feeding, and drinking) by extracting eight features from the accelerometer data and training KNN and SVM classifiers.

Adler et al. [3] used an accelerometer to estimate activity in two ways: a classification into active/inactive periods and a regression of the numeric “observed activity” depending on which activities are performed during a time window of 1 min. Both predictions are carried out based on a correlation between the input and the output, and no model is trained.

In [7], an accelerometer was used to measure 30 min of data from nine chickens, which were submitted to three degrees of poisoning. After applying a Kalman filter to remove noise, 18 features were extracted using FFT and the acceleration magnitude, and six activities were labeled from video footage and grouped into dynamic and static. Finally, five classifiers were trained to predict the poisoning treatment, and the correlation between activity frequencies and poisoning level was reported. The paper also reported that no differences were observed between the activities of chickens wearing an accelerometer and those who were not wearing it.

In [2], accelerometers were used to estimate the activity level (as the fraction of active time) through the thresholding of the acceleration after noise reduction. The authors observed some differences between chickens; the presence of the accelerometer had a tendency to improve the gait score and increase the frequency of preening. Then, they fit a univariate generalized linear model to estimate the gait score from the activity level and the weight, but only small effects were found.

These studies showed that IMU sensors have been used to monitor poultry behavior, effectively classifying activities such as feeding, resting, and walking. Machine learning models like KNN and SVM performed well in behavior prediction, while LSTM models were less effective. Studies have also shown that broilers are inactive for 78–80 percent of the day. Although these studies provide a solid starting ground and identified the potential of this technology, they were restricted to specific tasks, and none of them studied long-term attachments or continuous 24 h/day measurements of broilers between the attachment and the end of their lifetime.

#### 2.2.2. UWB and RFID

Both UWB and RFID are used to determine the position of broilers. However, the former can provide continuous exact-position updates, whereas the latter only provides broad indications of when an animal is present in a general zone.

Van Der Sluis et al. [15] tagged 12 chickens and recorded their location with UWB, making one measurement each 6.91 s and applying TDoA. They compared the distances estimated from the UWB sensor with that measured by video tracking and used these distances as an estimate of the activity level of the chickens. The recordings spanned 19 days, approximately 1 h/day, and they estimated that the error in the distance was less than 20 cm, in 85% of the measurements. This study was further extended in [22] to include RFID location. Giersberg et al. [5] carried out a similar study, using UWBs to estimate and compare the distance moved by 15 chickens hatched in three different systems.

Baxter and O’Connell [14] attached UWB backpacks to 30 chickens and observed them for focal periods of 15 min with a frequency of 10 Hz, filtering to remove speeds higher than 1 m/s with a Kalman filter. They observed that 85% of the positions were measured with an error of less than 50 cm and that there were no differences in the behavior of tagged and untagged broilers, except for the preening activity in week 6.

In [31], RFID tags were attached to laying hens’ legs to quantify the proportion of time spent feeding, drinking, and nesting. Other studies have also included RFID, but the technology can only detect when a chicken enters a certain area, limiting its usefulness for detecting low-granularity behavior. Therefore, it can be used as a complement to UWB to improve the location estimation or to measure the amount of time spent at different pen areas, but not for other aspects of welfare modeling, such as activity recognition.

In conclusion, UWB and RFID have been used to track the positions of broilers. UWB offers high accuracy with reported positioning errors of less than 20 cm. In contrast, RFID systems result in broader zone-based tracking and are more suitable for the monitoring of time spent by chickens in specific areas. It lacks the resolution necessary for detailed behavior analysis. RFID is more effective when used together with UWB for improved position tracking. Similarly to the previous work on IMU, these studies have yet to address the long-term analysis of continuously monitored data or the synergy between IMU and UWB technologies.

## 3. Experimental Settings

In this paper, we provide practical insights into the feasibility of using wearables for the long-term monitoring of broilers. To this end, we evaluated the efficacy of different attachment methods for wearable devices on broilers and their impact on welfare. Furthermore, we conducted the first research in the scientific literature that combines UWB and IMU sensors. This section describes the experimental setting that was followed for this study.

### 3.1. Animal Trials and Attachment Methods

The experiment was approved by the Ethical Committee of Poulpharm BVBA, Izegem, Belgium, with approval number LA1400564. Two rounds of the animal experiment were carried out according to the approved guidelines.

In the first round of the pilot study, ten one-day-old male Ross 308 chicks were housed for six weeks in a pen with 2 m × 1 m dimensions at Poulpharm. Out of the ten chicks, eight were attached with sensor tags (which combined UWB and IMU sensors within a single device) at least once during their lifetime, while two broilers were not subjected to any attachment. This study tested various attachment methods, including straps, sutures, 3D printed backpacks, tissue glue, eye lash glue, superglue, and epoxy. The chicks were given a single circular hopper feeder and a water tube with five nipple drinkers.

In the second round of the pilot study, 13 one-day-old male Ross 308 chicks were reared for 47 days in a pen with dimensions of 3 m × 3 m at Poulpharm BVBA. Out of these, nine chicks were attached with sensors at least once during their lifetime, while four chicks never had any attachment. The second pilot study tested superglue, epoxy, soft harness backpacks, complete 3D printed backpacks, and a combination of neck tags and superglue. Feed was supplied through two circular hopper feeders, and water was provided through a tube with nine nipple drinkers.

A commercial diet was provided to the chicks, with starter feed in the first two weeks, grower feed from the start of the third week to the fifth week, and finisher feed from the sixth week onward. Water and feed were provided ad libitum during the whole experiment. The environmental temperature was reduced by 1 °C every two days from 34 °C on day 1 until 22 °C was achieved, providing optimum thermal comfort for the broilers. The lighting schedule comprised 16 h of light and 8 h of dark. The lights were turned on from midnight to 2 a.m. and from 6 a.m. to 8 a.m. The dark period lasted between 2 a.m. and 6 a.m. and from 8 p.m. to midnight.

### 3.2. Sensor Infrastructure

The sensor infrastructure is composed of three elements: the sensors, a server that collects the data, and the anchors that measure the UWB data. All these elements for our experimental setting were designed and installed by Lopos [32].

To ensure that the data from the IMU and UWB sensors on the wearable are synchronized, the sensor tag (Figure 1) contains both UWB and IMU sensors, along with a 200 mA LiPo battery that powers them both. The entire tag is coated with a heat shrink plastic wrap to make it impervious to humidity and dust in a farm environment (Figure 1). The total weight of each tag was 20 g.

The first round of the study was used to test and calibrate the sensors. Then, in the second round, the measurements from the six sensors were systematically collected to carry out the analysis shown in Section 5 and Section 6.

For experiments where researchers will only utilize either UWB or IMU data, an integrated circuit is not beneficial. However, when researchers are interested in both sensors, there are several advantages to having both sensors on the same device. First, it makes for a lighter total weight, which is easier to attach to the animal. Second, a single battery can power both sensors. Third, both devices record data using the same clock, eliminating synchronization errors and difficulties derived from clock drift in offline systems. Furthermore, because all sensors of all chickens are uploading data to the same server, the clocks of the wearable devices are synchronized. Should this not be the case, offline devices can have a clock drift ranging up to a few seconds per day, which adds up to several minutes along the lifetime of a broiler. In this case, the data have to be manually synchronized, by periodically shaking the devices (to create a clear signal in the accelerometer) at specified times.

### 3.3. Noise Measurement

A complementary experiment was carried out to test the noise of the UWB and IMU sensors. A total of 80 devices were left motionless on the floor for 24 h and their UWB and IMU signals (at 1/16 Hz and 20 Hz, respectively) were monitored. This was performed simultaneously in four pens of 9×4 m each, with each pen equipped with six anchors.

## 4. Attachment Methods

### 4.1. Materials and Methods

The tags were attached to the broilers at different ages with the initial attachment at the start of the third week of age (Table 2). The weight of the attachment ranged between 21 and 42 g, varying with the method employed. Birds were individually restrained for sensor attachment. Retention periods for each method were recorded until detachment, removal due to injury or discomfort, or at the trial’s end on day 43. In case of detachments, the sensors were re-attached to the broilers with a fresh retention clock. Due to multiple detachments, a single broiler was often subjected to different attachment methods across its lifespan. The weight of the birds was recorded weekly. The birds were colored differently for individual identification, and they were regularly checked for any signs of injury, discomfort, or mortality. All different attachment methods are shown in Figure 2.

Each attachment method was tested on a minimum of two birds. Methods with fewer detachments, such as sutures, the combination of neck tag and superglue, and harnesses, had a small number of applications. Adhesive-based methods offered the potential advantage of reducing the attachment weight, which could facilitate easier carry by birds. Therefore, these methods, particularly those that use superglue and tissue glue, were subjected to additional repetitions following instances of detachment.

The variation in the age at which the attachments were initially applied was mainly influenced by the weight of the attachment package. Methods involving lighter attachments, such as most adhesive-based approaches, were initially applied during the third week of age. In contrast, methods that required heavier attachments, such as harnesses and 3D printed backpacks, were used in older birds. This pilot study involved a small sample size with the objective of obtaining preliminary information on the retention of each attachment method to identify promising approaches for further evaluation on a larger scale.

#### 4.1.1. Sutures

Sutures were used to attach UWB tags encapsulated in shrink-wrapped plastic. The tags were fixed on the chicken’s back using two stitches, 5 cm apart from each other. The first stitch was placed in the interscapular area near the neck and the second one toward the back. After tying the knot, the sensor was attached using the leftover thread of each stitch, leaving 4 cm of thread on each side. The ends of the thread were then tied together. This procedure was designed to avoid damage to the tissue around the stitches as the chicken grows. The extra thread was protected with tape to avoid other chickens interacting with it. Before suture placement, the chickens were treated with Midazolam intra-nasal (0.5 mg/kg) to provide sedation, anxiolysis, and mild analgesia.

#### 4.1.2. Adhesives

Different types of adhesives were tested: eyelash glue (essence lash glue, Kruidvat), tissue glue (Derma fuse 3 g, veterinary adhesive glue, mill pledge veterinary), superglue (Cyanoacrylate based), and epoxy glue. The adhesive was applied to the base of the sensor enclosure and the attachment site on the bird’s back. The sensor was then positioned and held in place for about four to five minutes to allow the glue to dry.

#### 4.1.3. Elastic Straps

The straps were made from white elastic material with a width of 1.2 cm. A knot was placed at the posterior end of the plastic coating of the sensor, with either side of the strap extending below the right and left wings. The strap at the anterior end passed through a hole made in the coating, where another knot was made to keep the length of the straps fixed to secure the attachment. Extra elastic was left to allow for length adjustment as the chickens grew.

#### 4.1.4. 3-D Printed Backpack

The sensor tag was glued on top of a 3-D printed backpack and attached with soft adjustable straps tightened around the wings of the broiler. An extra strap thread and adjustment buckle were enclosed in a small compartment below the plastic cover to which the sensor was attached.

#### 4.1.5. Harness

A small-size commercially available harness (Overstep the world store, AliExpress, China) made for chickens, duck, and goose was used for attachment. The anterior part of the harness, placed around the neck had a circumference of 24 cm, while the posterior part, with an adjustable belt and plastic buckle, was positioned on the chest and had a circumference 30–40 cm. The harness is primarily made of breathable mesh fabric. The wings of the broilers extended out from the opening on both sides. The buckle was used to loosen the backpacks as the broilers gained weight. The sensor was positioned on the back and was fixed with superglue on top of the harness.

#### 4.1.6. Neck Tag and Superglue Combination

The anterior portion of the sensor tag’s plastic wrapping was neck-tagged to the broilers. A tagging gun, typically used for attaching neck tags to birds for individual identification, was employed. A tagging clip passed through a hole in the wrapping and went across the loose layers of the skin around the neck. Due to the absence of loose skin on the back, the base of the sensor enclosure was superglued to the bird’s back for firm attachment and to reduce strain on the neck.

### 4.2. Results

#### 4.2.1. Attachment Methods and Retention Periods

Table 3 and Figure 3 display the time that the sensors remained attached to the chickens for each attachment method. Complementarily, Table 3 also lists the observed pros and cons of each attachment method. The following subsections provide a more detailed analysis of the observations for each attachment method.

#### 4.2.2. Adhesive-Based Methods

Glue-based methods showed varying degrees of success. Epoxy glue resulted in the longest mean retention time (7.5 days) among adhesives, followed closely by tissue glue (6.8 days) and superglue (6.17 days). Eyelash glue, although having non-irritant properties, showed the shortest retention period of 3 days. Tissue glue, already used in veterinary applications, did not cause observable stress during application. In contrast, superglue (Cyanoacrylate-based) application produced brief smoke and distress calls from the birds, indicating discomfort possibly due to heat generation upon skin contact. The early detachments of the glued sensors could be due to the regeneration of the skin at the contact site.

#### 4.2.3. Harness-Based Methods

The fabric harness showed the longest non-invasive retention period (13 days), but it had limited room for customization to fit at different ages of the bird. In the first round of the study, 3D printed backpacks were used on two birds, with one detaching after 2 days and another remaining attached until the bird’s death at 34 days. In both rounds, the 3D printed backpacks had an average retention of 9.5 days. Elastic band backpacks were removed after 3 days due to wing injuries, possibly caused by the poor design of the straps and the material of the elastic bands.

#### 4.2.4. Other Methods

The neck tag–superglue combination resulted in a consistent attachment of the anterior portion of the sensor. However, the detachment of the glued segment resulted in the sensor hanging from the neck, potentially causing discomfort and reduced data accuracy. This method required re-gluing after approximately every five days. Sutures, while providing long-lasting attachment, raise welfare concerns due to their invasive nature.

### 4.3. Observations on Bird Behavior and Welfare

Immediately post attachment, birds across all methods exhibited stress behaviors, including vigorous body movements, pecking at attachments, and moving backward with a lower posture. These behaviors were most pronounced with strap-based backpacks. In the first 1–3 days following attachment, birds showed lower posture walking and a lower tendency to locomote.

## 5. Ultra-Wideband (UWB)

### 5.1. Materials and Methods

The UWB infrastructure comprises stationary UWB anchors and mobile UWB tags attached to the broilers. TDOA was used to estimate the locations. Four UWB anchors were installed around each pen, mounted directly on the fences at approximately 30 cm above the floor. For optimal performance, anchor placement should aim to be coplanar and non-colinear with the tags, as suggested in [33]. This configuration can be easily achieved by positioning the anchors at the pen’s boundaries and near the corners, maintaining a consistent height above 30 cm (to mitigate first-path propagation coincident reflections).

The small size of the battery imposed a strong limitation on the volume of data that could be collected. To enable the continuous monitoring of the IMU signal along the entire lifetime of the broilers (after attachment in week 3), the UWB sampling rate was set to one measurement every 32 s. This is the result of a trade-off: while lowering the sampling rate extends battery life, it also reduces localization precision. Since broilers can move back and forth within a span of a few seconds, reducing the sampling rate might lead to underestimating the distance they travel during the experiments.

### 5.2. Results

#### 5.2.1. Noise Analysis

To analyze the noise of the UWB position estimation, TODO individual tags were placed on predetermined locations in the pen. Figure 4 plots the reported location of each of these static tags during 24 h, illustrating how the errors differ in the center of the pen versus at the edges of the covered area. Tags that are located close to the corners of the pen (i.e., close to the anchors) show a noise distribution in which the X and Y coordinates are correlated. Tags around the center of the pen show round shapes, indicating a smaller correlation between X and Y. This reflects the functioning of UWB localization, which solves a set of equations based on the time it takes for a signal to travel between the tag and the anchors, as described in Section 2.1. When a tag is close to the anchors, the crossing point of the hyperbolic curves drawn by each anchor is more difficult to compute, as the directions of the hyperboles are more similar to one another, causing the noise to have a hyperbolic-like shape. As such, animals in the center of the pen will have both lower errors on both X and Y coordinates. One way to remedy this in future experiments is to move the installed anchor nodes further away from the borders of the pen, thereby reducing the uncertainty (also referred to as the “dillution of precision”) of the position of the tags close the anchor nodes.

Figure 5 shows the 95% confidence interval for the horizontal and vertical axes for each tag, assuming a Gaussian distribution. It is clear that the magnitude of the noise varies between positions.

#### 5.2.2. Areas Frequently Visited by the Chickens

Figure 6 shows the frequency of occupation of different areas in the pen. A Gaussian filter was applied on the 2-D histogram (with σ=3 to account for the noise in the UWB sensor). Note how the two feeders are clearly drawn in the center of the pen, and the drinkers on the left, indicating that the chickens spend most of their time around them.

However, when the chickens are considered individually, they appear to exhibit different behaviors. Figure 6 reflects this expectation: although the chickens generally move in the same areas, some of them prefer some areas more than others. For instance, the chicken with tag 128 (resp. 21) spent more time in the left (resp. right) part of the pen. This observation reflects the need for individual chicken monitoring to analyze behaviors related to the chicken location, such as changes in feeding or drinking habits, in walked distance, or in interaction with other chickens.

#### 5.2.3. Movement Variability Between Chickens

Although UWB technology is primarily designed for location, by measuring the Euclidean distance between two consecutive measurements, we can estimate the minimal distance walked by each chicken in the 32 s elapsed. Two types of errors are involved in this measurement: on the one hand, an underestimation caused by the limited frequency of the measurement; on the other hand, an overestimation caused by the accumulation of the measured noise. Figure 7 shows the estimated distance walked by each chicken along the experiment every 4 h. We can see some small differences in the distance walked by each chicken, although there is no clear trend in the pattern along the weeks that were measured.

## 6. Inertial Measurement Unit (IMU)

### 6.1. Materials and Methods

The sampling frequency is the most important hyperparameter for this device. A low sampling frequency allows for a longer battery time, but some activities might not be detected. On the other hand, a high frequency enables the capture of high-resolution data, but has a higher battery consumption. For this experiment, the accelerometer was sampled at 20 Hz (3-axis), which was high enough to detect activities in different animals in previous research [7].

Broilers spend most of their time in an inactive state, either resting motionless or sleeping [4]. Therefore, the first step in activity recognition often consists of distinguishing between active and inactive states. When using an IMU device, it is common practice [3] to compute the magnitude of the acceleration according to Equation (Equation 1). When the sensor is motionless, the magnitude of the acceleration vector is 1 g (equal to the force of gravity, pointing downward). Motions of the sensor induce deviations from this value; therefore, by establishing a threshold on these deviations, it is possible to classify time periods into active or inactive.(1)$$m=\sqrt{x^{2}+y^{2}+z^{2}},$$

### 6.2. Results

In this section, we will examine activity levels among different chickens, measured using IMUs. During the second round of experiments, each chicken was outfitted with an IMU sensor that can capture acceleration in the *x*, *y*, and *z* directions, sampled at a rate of 20 Hz, leading to a total of around $1.36\times 10^{9}$ measurements.

We obtained the magnitude of the acceleration for every measured time point. Figure 8 shows the distribution of the acceleration in the three axes along the entire round for each chicken. It clearly shows how the orientation of each sensor can be significantly different between chickens, which can be due both to differences when attaching the sensor (despite the care taken in doing so) and to differences in chicken resting positions and behavior. The magnitude of the acceleration is a much more robust and chicken-independent measure of the activity, as shown in Figure 9. However, small differences in distribution are still observed between chickens, stressing the importance of adequate normalization to reduce these differences before attempting to train an activity detection classifier. The plot also shows that the standard deviation of the magnitude is low, meaning that most values correspond to inactive times.

The choice of a threshold to distinguish between active and inactive times will determine the balance between false positives and negatives in activity detection. Figure 10 shows the proportion of the data that would be detected as active for each possible threshold. As we can see, the distribution is unimodal: there is no clear separation between active and inactive times; therefore, the selection of the threshold can be a critical decision. In previous research [2], a threshold of 0.05 g was used; in our data, this would mark 5.04% of our total accelerometer measurement data as active. After filtering the data and selecting all 1 s periods with at least some activity in them, we marked 23.78% of the data as active, as shown in Table 4. Complementarily, Figure 11 shows the percentage of active time for each day and chicken.

## 7. Combining IMU and UWB Sensors in a Single Wearable

The UWB and IMU sensors provide different insights, namely position data and activity levels, and as such, they provide complementary data. However, they both consume energy from the battery and increase the total weight of the wearable. Therefore, in this section, we evaluate the synergy between these sensors, determining whether they complement each other or exhibit redundancy in their data.

### 7.1. Activity at Different Locations

Figure 12 shows different statistical measures of the acceleration magnitude recorded, on average, at each location of the pen for each chicken. It can be observed that the maximum and standard deviation of the acceleration recorded are good indicators of the activity level. Furthermore, we can see higher activity levels around the feeders and the drinkers, although this pattern differs greatly between different chickens. Based on this histogram, it can be observed that even when broilers do not actively move to a different location, they can still exhibit high activities in their locality. This information would be lost when considering only position data.

### 7.2. Movement Correlation with IMU

Since the cost of an IMU wearable is lower than the cost of IMU sensors, it is worth investigating if the movement of the animals could be estimated using the more low-cost sensors. To this end, here, we analyze the correlation between UWB movement and IMU data. With a high correlation (up to 1 or up to −1), we can estimate how much chickens physically move (which is not always equal to activity levels) based only on IMU sensor information, without relying on a high UWB update rate. With a low correlation, such a prediction would not be possible. Figure 13 presents such correlations with applied preprocessing. The labels on the *y*-axis denote these preprocessing steps (variables). The first variable (16S, 1T, 5T, 10T) indicates how much both UWB and IMU data are resampled (16 s or 1, 5, or 10 min). IMU information is always first resampled using an average of 1 s, and the resampling variable is denoted on the *y*-axis label. The second variable corresponds to which axis of the IMU data is used for resampling. UWB always calculates the traveled (Euclidean) distance over the aforementioned resampling period. The last variable indicates whether the mean or standard deviation (std) of the IMU data is used for resampling. Once both IMU and UWB data are resampled to the same time window (using the preprocessing steps mentioned above), a Pearson correlation is measured between the two. As a result, the highest correlation (0.33) is obtained for all chickens using a (IMU) resampling period of 5 min, the IMU *x*-axis, and the standard deviation. Interestingly, a longer window of 10 min yields only a slightly lower correlation, while the magnitude of the IMU signal sits just behind, on the third and fourth highest correlations. Lower resampling periods (16 s, matching the default UWB update rate) yield very poor correlations. Although the highest correlation is not near 1 (or −1), clearly, some correlation can be found, indicating that IMU data can be viable (when using the standard deviation over a long period) to indicate whether the chicken has moved in space or sat still.

### 7.3. Sitting Still or Activity (IMU) Correlation with UWB

The energy consumption of transmitting a UWB packet for localization is significantly higher than the local storage of sensor data using on-chip memory. As such, intelligently sending localization packets can significantly increase battery lifetime and/or reduce the weight of the wearable. Here, we analyze the IMU activity for chickens not physically moving through the pen (traveling at least distance X). This will indicate whether it is possible to trigger UWB localization only when the chicken is moving based on IMU activity data. The hypothesis is that we can differentiate the positional movement versus stationary movement of the chicken using IMU sensors. Figure 14 shows the distribution of distances traveled by all chickens in 30 min. The distance distribution is shown for IMU sensors that had low or high movements, orange and blue, respectively. The IMU data (magnitude) are grouped by the sum of standard deviation values of 1 s over half an hour. The UWB data are calculated by grouping the sum of distances over half an hour. Low IMU movements (experimentally derived based on the lower side of the IMU distribution) had movements lower than 5 m in half an hour. If a threshold is set based on the sum of IMU movement in half an hour, a high accuracy (i.e., the chickens start moving in space) will be achieved with only a few false-negative movements, enabling the UWB to sleep until IMU movement is detected.

## 8. Overall Guidelines and Lessons Learned

In the previous sections, we presented the results obtained in two rounds of experiments with several chickens wearing a UWB and IMU sensor continuously for 3 weeks, and we compared several attachment methods for these sensors. In this section, we further discuss the benefits and challenges of these wearables.

### 8.1. Considerations for Sensor Attachment

#### 8.1.1. Age and Weight Considerations

Following established guidelines, the combined weight of the sensor and associated attachment accessories should not exceed 5% of the bird’s body mass [18,19]. The additional weight of the wearable device may cause stress and discomfort to the animals. Delaying attachment to later stages of growth is advisable, as increased body mass and strength in older birds may better accommodate on-body attachments. The primary attachment site, located on the backbone between the wings, would be suitable for smaller, vertically oriented sensor designs. Sensor development and attachment methods should consider the anatomical features of broilers to minimize interference with the bird’s natural movements and behaviors.

#### 8.1.2. Attachment Methods

This pilot study reveals the challenges of attaching on-body sensors to broiler chickens. While adhesive-based methods indicate minimal invasiveness but lower retention, harness-based methods provide longer retention but present challenges of causing injury and the need for regular readjustments. The sutures and neck tag–superglue combination shows promise but requires refinement to reduce the invasiveness and need for re-application. We suggest the need for a multi-dimensional approach to select the optimal sensor attachment method that considers retention time, data quality, behavioral impact, and animal welfare.

#### 8.1.3. Future Research

We recommend further studies to investigate the following aspects:Explore novel attachment methods and materials that may offer improved retention without affecting the bird’s welfare, behavior, or physical integrity. The mode of attachment of activity monitors affects data quality [34]. Loosely fitted sensors, particularly accelerometers, may lead to noisy data and the overestimation of activity levels. Future work should quantify these effects. In addition, it would be beneficial to develop age-specific attachment solutions to account for rapid broiler growth.Quantify the effects of different attachment methods on bird behavior and welfare. Future studies should systematically assess the effects of each attachment method on walking ability, wing health, bird posture, access to food and water, social behaviors, and vulnerability to the pecking and harvesting of red mites.Impact on appearance and social interactions. The attachment of the tag may influence the broiler’s natural behaviors, such as foraging, dust bathing, and social interactions. Natural behaviors are important indicators of good welfare. The attachment also results in an altered bird appearance, which could potentially affect the flock dynamics. Further research is needed to quantify these effects and develop designs that minimize disruption to natural social interactions.

These insights provide a foundation to develop more effective and welfare-friendly sensor attachment methods for broiler monitoring, which could potentially lead to improved welfare assessment and management practices in poultry research.

### 8.2. Considerations of Sensor Technologies

The results shown above demonstrate the information that can be retrieved from the different sensors.

#### 8.2.1. UWB Considerations

The UWB sensor allowed us to locate the chickens at all times. Unlike video-based approaches, UWB sensors do not depend on the lighting of the room and are robust to occlusions. Although modern video monitoring systems such as infrared and thermal imaging perform well in low-light conditions, they are less reliable for tracking individual animals in group settings and are more impacted by visual obstructions than UWB sensors. This highlights UWB as a very suitable technology for monitoring chickens continuously, both day and night and in high-density commercial settings where individual chickens are difficult to track in a video. Furthermore, it was possible to estimate the distance walked by each chicken based on the UWB locations, which further allowed us to identify differences in the distance walked by different chickens. This methodology has the potential of detecting health issues in chickens (in particular, lameness), when the distance walked by a chicken is significantly lower than that of other chickens, or of the same chicken in a previous period of time.

#### 8.2.2. IMU Considerations

Complementarily, the IMU sensor allowed us to detect the activity of the chickens. The raw three-axis signal is strongly dependent on the individual chicken, due to differences in the sensor attachment and the chicken position. However, the magnitude of the acceleration showed a more robust signal, albeit still showing some differences between individuals. It would be necessary to consider both the raw three-axial signal and its magnitude to identify complex activities in the chickens; however, this would necessitate adequate preprocessing to eliminate differences coming from individual chickens.

#### 8.2.3. Future Research

Despite the promise of this technology, there are still several technical challenges that must be addressed before it can be deployed in large-scale commercial farms:**Battery weight and duration**. To maintain the weight of the wearable under 5% of the chicken’s body weight, it is necessary to minimize the size of the battery, which incurs power restrictions to the device. In this experiment, we were limited to measuring the location of each chicken every 32 s; however, a higher update rate of the UWB would lead to a more accurate distance estimation.**Bandwidth minimization**. Due to the small scale of this experiment, it was possible to wirelessly monitor six chickens continuously with a 20 Hz frequency for the IMU sensor, whereby all sensor data were wirelessly transmitted. However, in real situations where more chickens are monitored, the network can become saturated by the large amounts of data that must be read each second, coming from tens of sensors simultaneously, requiring either the on-chip processing or on-chip storage of data.The required throughput for transmitting IMU data will depend on (i) the sampling frequency of the IMU, (ii) the sample size of each IMU measurements, and (iii) the number of monitored animals. Increasing the sample frequency and sampling size will result in higher accuracy measurements at the cost of additional network traffic. In our experiments, IMU data were captured at 20 Hz, using three axes for each 8 bit per IMU value, resulting in 60 bytes/second/chicken. We transmitted the IMU data wirelessly using the license-free 915 Mhz ISM band. The range between two devices using subGHz is in the order of 1000 m (open area). In case more data need to be transmitted, it is possible to instead use technologies with higher bandwidth, such as WiFi.**Noise reduction**. It was seen in Section 5.2.1 that the noise in the UWB measurements is not negligible and that it can be mitigated by optimizing the location of the anchors. Further research can also be carried out about specific preprocessing algorithms that reduce or eliminate this noise.**Energy management.** To utilize vibrational energy, kinetic energy harvesters can be used. For harvesting energy from machinery, these are very efficient: up to 250 mW for kinetic energy harvesting on trains. When harvesting energy from human body movement, the energy is significantly lower, in the order of 10 uW up to 100 uW, depending on the movement type (walking versus running). This is several magnitudes lower than the energy availability required to continuously transmit data. There are no scientific publications on energy availability when using kinetic energy harvesters on chickens, but this is expected to be significantly less then when using human activity energy harvesting, further reducing the feasiblity of using vibration-based energy collection for these purposes.

These challenges lead to a subsequent research line: the development of embedded algorithms that continuously monitor the signal read by the device and only upload that information to the server when a relevant change or activity is detected. Although libraries and models for embedded machine learning exist, developing a model that can run on very small batteries can prove to be a very challenging task. Furthermore, such an approach would also solve the problem of limited bandwidth for large-scale settings.

## 9. Conclusions

In this paper, we evaluated various attachment methods for applying wearable devices on broilers, including UWB and IMU measurements. We tested the typical noise that can be expected in the measurements, and we continuously monitored the distance moved and acceleration of the broilers from week 3 to the time of death, identifying typical patterns in both sensors. Furthermore, we established the relation that can be expected in the data collected simultaneously from UWB and IMU sensors.

Based on our observations, we established a set of guidelines on which sensors to use for specific use cases for practitioners in the field. A summary mapping the best suitable sensors for different research purposes is shown in Table 5.

To conclude this paper, we identified several technical and welfare challenges that limit the current applicability of wearable devices on broilers, and we proposed potential research lines that are open for tackling such challenges.

## Figures and Tables

**Figure 1 sensors-25-00811-f001:**
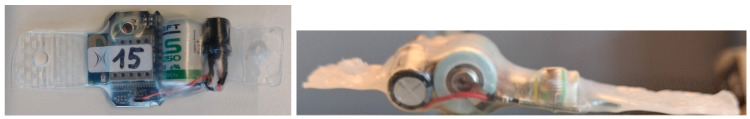
Tag containing UWB, IMU, and battery.

**Figure 2 sensors-25-00811-f002:**
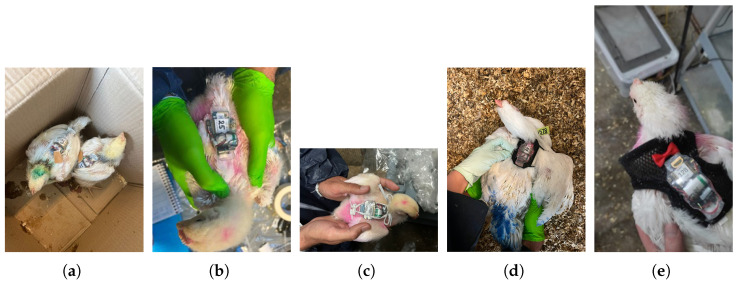
Attachment methods tested in this study. (**a**) Sutures. (**b**) Adhesives. (**c**) Elastic straps. (**d**) Three-dimensional printed backpack. (**e**) Harness.

**Figure 3 sensors-25-00811-f003:**
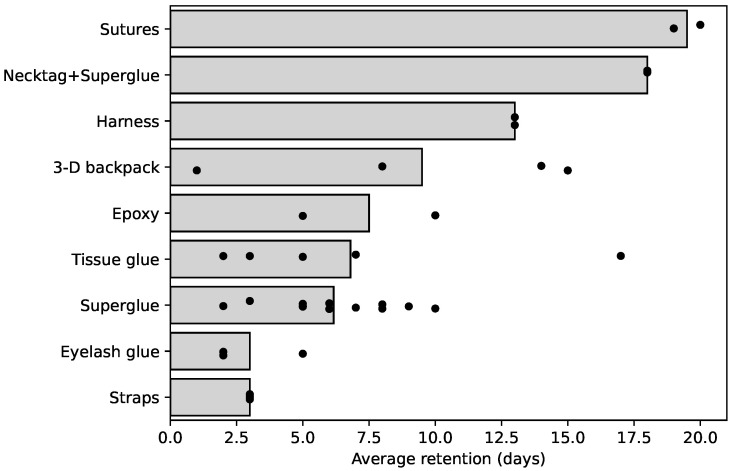
Mean retention time for each retention method. Dots indicate individual measurements.

**Figure 4 sensors-25-00811-f004:**
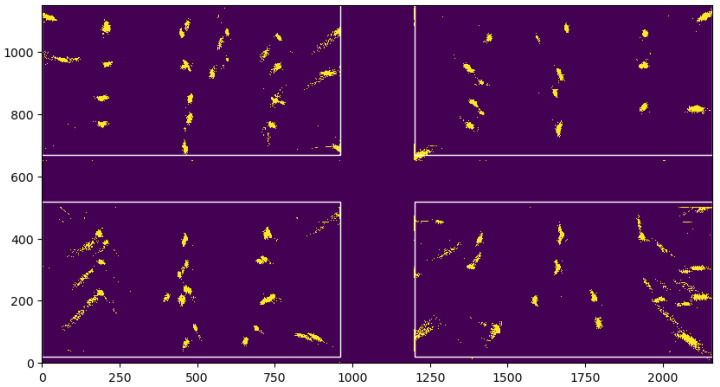
Frequency of measurement at each location in the four pens of the noise measurement experiment. Each colored blob corresponds to one UWB tag that was left on the ground for 24 h; the area size indicates the magnitude of the noise. The units of measurement on both the *x*-axis and *y*-axis are in centimeters.

**Figure 5 sensors-25-00811-f005:**
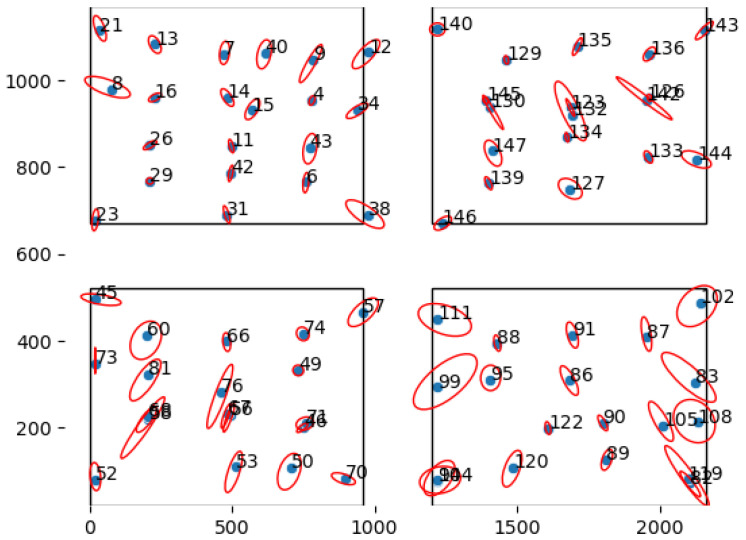
Blue points mark the average locations measured for the tags; the red ellipses mark their 95% confidence interval (assuming Gaussian distribution).

**Figure 6 sensors-25-00811-f006:**
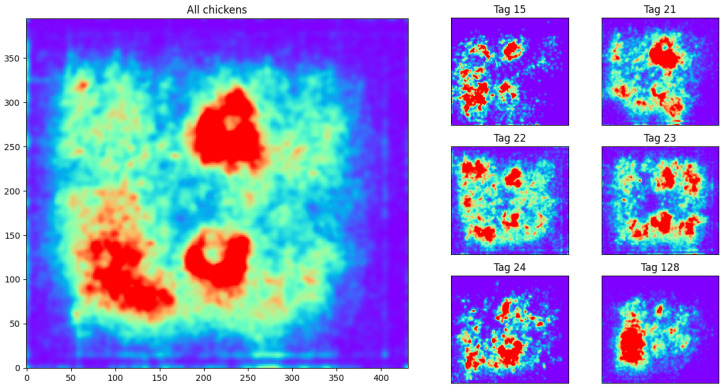
Heatmaps of chicken locations in the pen. The left plot aggregates data from all chickens; on the right, a separate heatmap is depicted for each individual chicken. Red areas indicate frequent occupation.

**Figure 7 sensors-25-00811-f007:**
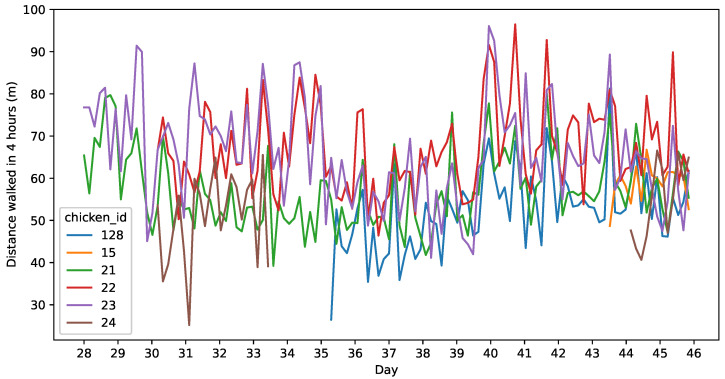
Estimated distance walked every 4 h by each chicken. The data are from the second round of the experiment.

**Figure 8 sensors-25-00811-f008:**
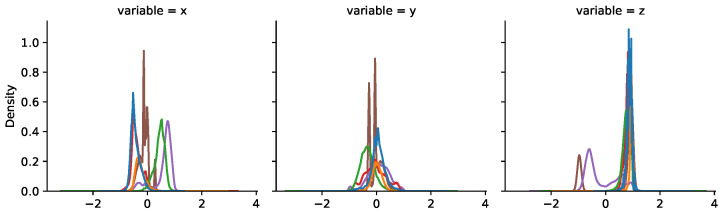
Distribution of accelerations measured in the three axes for each chicken.

**Figure 9 sensors-25-00811-f009:**
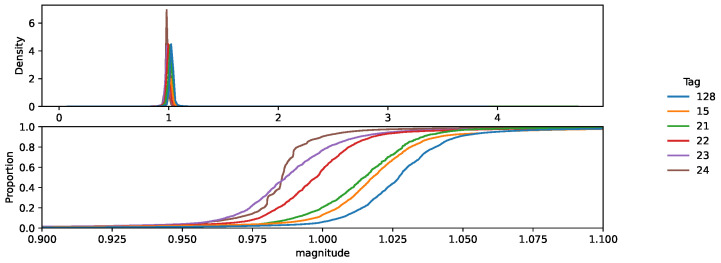
Distribution and cumulated probability (zoomed) of the acceleration magnitude for each chicken before normalization.

**Figure 10 sensors-25-00811-f010:**
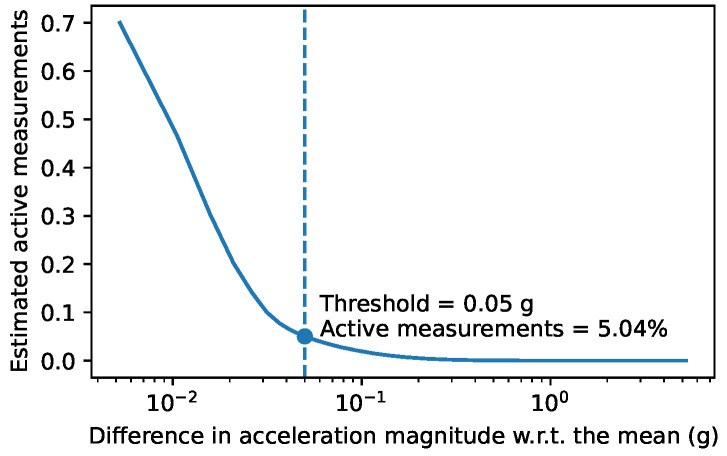
Cumulative distribution of the difference between the acceleration magnitude and its mean.

**Figure 11 sensors-25-00811-f011:**
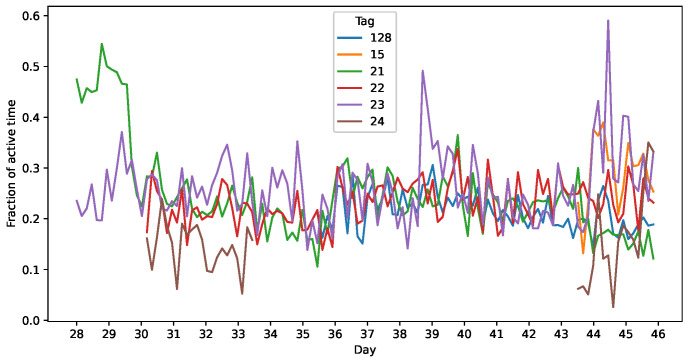
Proportion of active time in 4 h intervals for each chicken.

**Figure 12 sensors-25-00811-f012:**
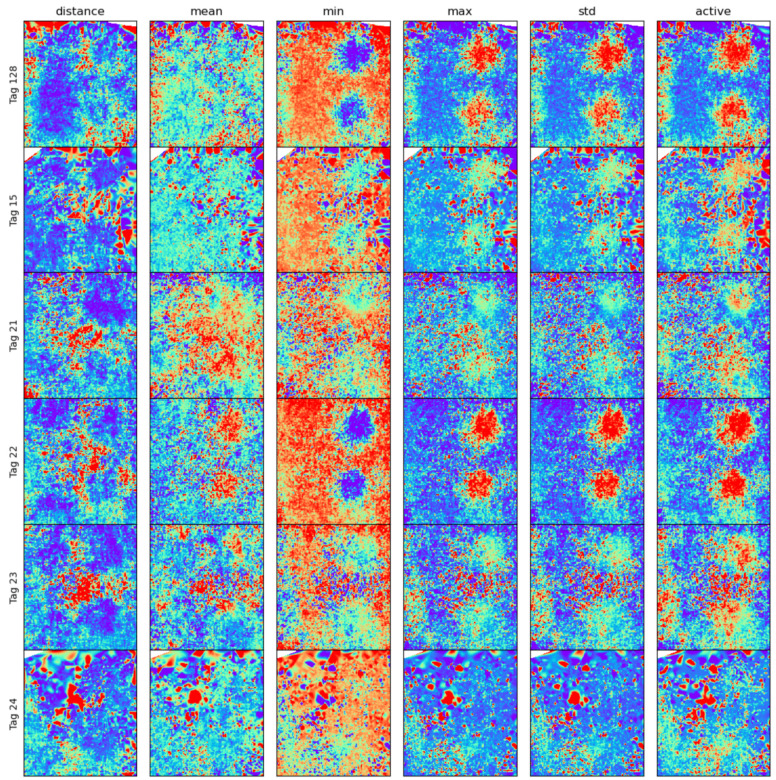
Histogram of activity measures at each location of the pen for each chicken. Red indicates higher values.

**Figure 13 sensors-25-00811-f013:**
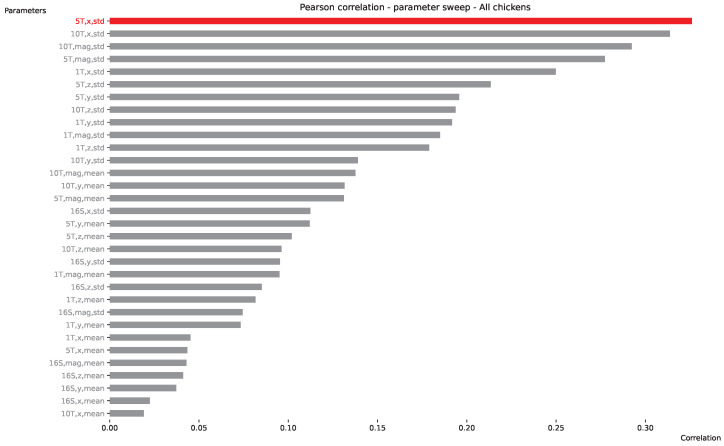
Correlation movement vs. IMU.

**Figure 14 sensors-25-00811-f014:**
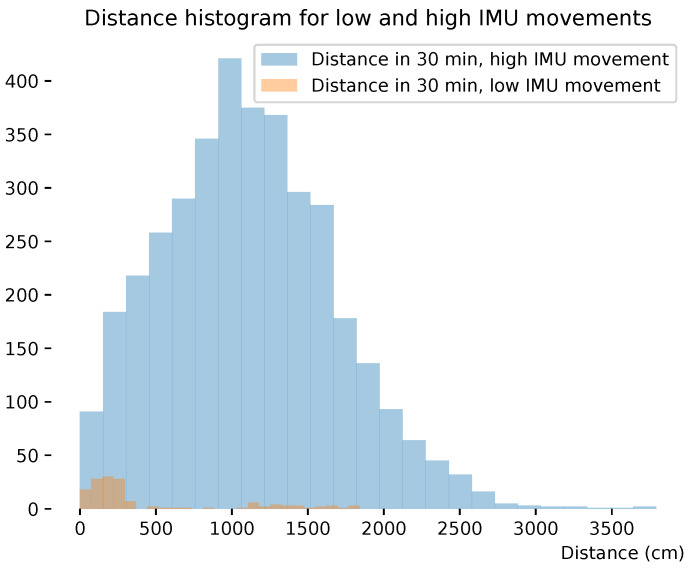
Histogram IMU vs. movements.

**Table 1 sensors-25-00811-t001:** Overview of previous work using wearable devices on poultry.

Reference	Year	Animals	Research Goal	Number of Chickens	Wearable Devices	Multiple Attachments	Update Rate
[15]	2019	Broilers	Distance estimation	12	UWB	Elastic bands around wing base	1/6.91 s
[22]	2020	Broilers	Distance estimation	34	UWB + RFID	Rubber bands to the legs	13.56 Hz
[14]	2020	Broilers	Distance estimation	27	UWB	Backpack	10 Hz (after pilot trial)
[5]	2023	Broilers	Distance estimation	15	UWB	Elastic bands around wing base	2 Hz
[8,9]	2018	Broilers	Activity classification	-	Accelerometer	Backpack	100 Hz
[8]	2018	Broilers	Activity classification	-	Accelerometer	Backpack	100 Hz
[10]	2021	Broilers	Activity classification	9	Accelerometer	Harness	40 Hz
[4]	2021	Broilers	Activity detection	280	Accelerometer	Backpack	0.35 to 3.5 Hz
[3]	2023	Broilers	Activity detection	5	Accelerometer	Elastic bands around wing base	-
[6]	2021	Broilers	Disease detection	24	Accelerometer	-	-
[2]	2023	Broilers	Lameness detection	15	Accelerometer	Adhesive tape	100 Hz
[7]	2023	Broilers	Poisoning detection	9	Accelerometer	Harness	20 Hz
[29]	2012	Laying hens	Activity classification	6	Accelerometer	Harness (after experiments)	10 Hz
[12]	2023	Laying hens	Activity classification	4	Acc. + Gyroscope	Harness	1000 Hz
[11]	2014	Laying hens	Jump detection	6	Accelerometer	Harness (after experiments)	100 Hz

**Table 2 sensors-25-00811-t002:** Number of applications, number of animals, and age of initial attachment for each method (both rounds).

	Animals	Total Number	Sensor + Attachment	Age of Initial Attachment	
**Method**	**Tested**	**of Applications**	**Weight (g)**	**Days**	**Week**
Elastic straps	6	6	24	17	3rd
3-D Printed Backpack	4	4	34	21	4th
Sutures	2	2	22	22	4th
Tissue glue	4	5	21	14	3rd
Eye lash glue	2	3	21	25	4th
Superglue	7	12	21	14	3rd
Epoxy	2	2	21	14	3rd
Mesh Harness	2	2	42	29	5th
Neck tag + Superglue	2	2	22	29	5th

**Table 3 sensors-25-00811-t003:** Summary of the measured retention time (in days) from the tested attachment methods, and pros and cons of each method based on the observations in the study.

Method	Repetitions	Mean	Min	Max	Range	St. Dev.	Pros	Cons
Sutures	2	19.5	19	20	1	0.71	Long retention	Invasive
Neck tag + Superglue **	2	18	18	18	0	0	Long retention	Detachment of glued part of sensor
Fabric Harness **	2	13	13	13	0	0	Longest non-invasive retention	Limited customization
3-D backpack	4	9.5	1	15	14	6.45	Adjustable straps	Apparent discomfort
Epoxy glue	2	7.5	5	10	5	3.54	Longest retention among glues, lightweight attachment	Moderate retention
Tissue glue	5	6.8	2	17	15	6.02	No observed stress during application, lightweight attachment	Moderate retention
Superglue	12	6.17	2	10	8	2.37	Lightweight attachment	Results in distress during application
Eyelash glue	3	3	2	5	3	1.73	Non-irritant ingredients, lightweight attachment	Weak retention
Straps *	6	3	3	3	0	0	Easy application	Caused wing injuries

* Removed after 3 days because of observed injuries in the wings of the broilers. ** None of the tags fell within the observed period.

**Table 4 sensors-25-00811-t004:** Percentage of active and inactive time for each chicken with a threshold of 0.05 g.

Tag ID	Inactive	Active
128	78.47%	21.53%
15	70.88%	29.12%
21	77.06%	22.94%
22	75.63%	24.37%
23	73.12%	26.88%
24	84.75%	15.25%
Total	76.22%	23.78%

**Table 5 sensors-25-00811-t005:** Summary of this study’s findings.

Technology	Output	Research Questions
	Activity level estimation	
	Distance estimation	
UWB	Area/location analysis	Welfare evaluation
	Social interaction identification	Performance monitoring
	Activity level estimation	Disease detection
IMU	Motion detection	Poisoning detection
	Activity recognition	

## Data Availability

Data sharing requests can be addressed to us, but we can not promise we will be able to share the data.

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
