# Peer review of "Monitoring Welfare of Individual Broiler Chickens Using Ultra-Wideband and Inertial Measurement Unit Wearables"

_sensors, 2025, doi:10.3390/s25030811_

Round 1
Reviewer 1 Report
Comments and Suggestions for Authors
This paper explores the use of wearable devices (UWB and IMU sensors) for monitoring poultry welfare. However, there are several inconsistencies and lacking details in some sections that need to be addressed to improve clarity, scientific rigor, and technical feasibility.
- The data shown in Figure 7 is unclear regarding which group it represents. The paper mentions 10 chickens and 13 chickens in different experimental groups, but Figure 7 does not specify whether the data is from the first or second trial, or if it combines both. Clarify the data source in the figure legend or the text.
- In section 8.2.3, the authors mention data from 5 chickens, but earlier sections refer to 10 chickens and 13 chickens, with no clear explanation of where the 5 chickens' data comes from. Meanwhile, the subsequent experimental results were all based on the data of 6 chickens. Can the sample size represent the whole experiment? The authors should clarify and ensure the sample size is appropriately discussed.
- The paper does not clearly discuss the impact of sensor installation on group behavior, such as whether there was any rejection or aggression between the chickens wearing sensors (e.g., pecking or attacking).
- In section 8.2.1, the authors mention network saturation but do not provide specific load data, such as data transmission per second, network bandwidth usage, or actual utilization rates, making it difficult to assess the practical limits of the system. Provide more details on network load and utilization.
- The paper does not mention dynamic energy harvesting methods (e.g., vibration-based energy collection) to reduce the weight of the sensor and extend its operational lifetime. This method is highly relevant to solving the issues of battery weight and energy limitations.
- In section 8.2.3, the authors state that UWB does not depend on lighting, but this overlooks the capabilities of modern video monitoring technologies (e.g., infrared or thermal imaging), which can also work in low light conditions. A more balanced comparison between UWB and video technologies, especially in environments with low or no lighting, would be beneficial.
- The authors use metrics such as acceleration to assess chicken activity, but all chickens wore the devices. There was no control group without sensors to compare whether the activity levels are abnormal or if the devices themselves impact the behavior.
Author Response
Dear Reviewer 1,
Please see the attachment.
Kind regards

Reviewer 2 Report
Comments and Suggestions for Authors
Title: I suggest reducing the title to just:
Monitoring Welfare of Individual Broiler Chickens using UWB and IMU Wearables
Abstract: the authors should briefly address the main findings and conclusions in the abstract.
Introduction: it is very well founded, providing a good background to what is already known and adequately raising the gaps to be filled with the research in question.
L20-21: improve the sentence, it is repetitive. In just two lines, the word welfare was mentioned three times
In section 2.1, the authors provide a sufficient explanation of the operating principles of the wireless sensor technologies under evaluation, as well as their limitations in use.
In sections 2.1.1 and 2.1.2, the authors discuss the articles presented in table 1. I suggest mentioning the main conclusions and challenges reported by the researchers
In section 3, the authors should describe whether the experimental methods followed the ARRIVE guidelines and were approved by an ethics committee.
Section 3.1. describe in detail the methods of attaching the sensors to the birds, especially for superglue, epoxy, and a combination of neck tags and superglue, which seem to have the most impact from a welfare point of view.
Pg 6. L252-261. I believe this discussion is more pertinent in the initial contextualization topics.
Section 3.3. How was the noise measured?
L266-267 - exclude
Include the information on the pilot experiments in section 4. Material and Methods.
Why were different numbers of birds used for each fixation method? Just two birds for some of the methods is insufficient for a reliable answer.
Why was there variation in the age of fixation in the different methods? This is not clear.
L276 - “Due to multiple detachments, a single broiler was often subjected to different attachment methods across its lifespan”. Couldn't this have interfered with the birds' behavior?
Section 4.1.1 Sutures and Neck-tag and Superglue combination. No anesthesia was used for the procedures, only a soothing ?
Adhesives. Were any tests carried out for potential tissue damage (skin) due to the application of different types of glue? How long were the birds restrained until the glue dried?
Figure 4: What is the unit of measurement?
L383-387 - this seems to be a discussion and not results
Author Response
Dear reviewer 2,
Please see the attachment.
Kind regards,

Round 2
Reviewer 2 Report
Comments and Suggestions for Authors
All the suggestions and corrections were accepted by the authors or adequately justified. There was a significant improvement in the clarity of some aspects that were pointed out. I hope I have contributed to the refinement of the article. Congratulations on the research and success to the researchers